# The Anti-Cancer Potential of Heat-Killed *Lactobacillus brevis* KU15176 upon AGS Cell Lines through Intrinsic Apoptosis Pathway

**DOI:** 10.3390/ijms23084073

**Published:** 2022-04-07

**Authors:** Chang-Hoon Hwang, Na-Kyoung Lee, Hyun-Dong Paik

**Affiliations:** Department of Food Science and Biotechnology of Animal Resources, Konkuk University, Seoul 05029, Korea; fst94hun@naver.com (C.-H.H.); lnk11@konkuk.ac.kr (N.-K.L.)

**Keywords:** *Lactobacillus brevis*, probiotics, apoptosis, cancer therapy, caspases

## Abstract

Recent research has focused on the anti-cancer properties of *Lactobacillus* strains isolated from fermented foods. Their anti-cancer effects are caused by the apoptosis induction in cancer cells. However, sepsis, which can occur when cancer patients consume living organisms, can cause serious conditions in patients with reduced immunity because of cancer. Therefore, this study was conducted using heat-killed *Lactobacillus brevis* KU15176 (KU15176). To determine the relationship between inflammation and cancer, the anti-inflammatory effect of KU15176 was evaluated using a nitric oxide (NO) assay. Then, 3-[4,5-dimethylthiazol-2-yl]-2,5 diphenyl tetrazolium bromide (MTT) assay was conducted to select cancer cells that showed the anti-proliferative effect of KU15176. Next, reverse transcription-polymerase chain reaction (RT-PCR), 4′,6-diamidino-2-phenylindole (DAPI) staining, flow cytometry, and caspase colorimetric assay were performed. As a result, it was confirmed that KU15176 could cause the increasing expression of apoptosis-related genes (*Bax*, *caspase-3*, and *caspase-9*), DNA breakage, effective apoptosis rate, and increased caspase activity in the human stomach adenocarcinoma (AGS) gastric cancer cell line. In conclusion, these results suggest a potential prophylactic effect of KU15176 against cancer.

## 1. Introduction

Probiotics are living microorganisms that offer hosts various health advantages [1]. Lactic acid bacteria (LAB) are a common probiotic strain that is widely used in industry. Several studies have revealed that probiotics exhibit functional effects, such as anti-inflammatory [2,3], antioxidant [4,5], immune-enhancing [6,7], antimicrobial [8], antidiabetic activities [9], and the removal of heavy metal [10]. Recently, the study of anti-cancer activity has focused on protective adjuncts against a host of diseases [11]. Although there are various synthetic agents exhibiting anti-cancer activity, these agents are not only expensive, but doubts exist regarding their long-term safety and stability [12]. Numerous reports have indicated that probiotics and their metabolites exert anti-cancer effects [13,14]. However, the mechanisms by which probiotics exert their anti-cancer effects remain unknown.

Programed cell death, referred to as apoptosis, occurs normally as a homeostatic mechanism for maintaining the cell population in tissues [12]. Abnormal cell proliferation occurs in cancer cells due to functional mutations in genes associated with apoptosis; therefore, apoptosis is a major topic in non-surgical cancer treatment [15,16]. Thus, the anti-cancer effect was evaluated through changes in the expression of apoptosis-related genes (*Bax*, *Bcl-2*, *caspase-3*, and *caspase-9*). 

Unfortunately, the safety of live *Lactobacillus* strain intake has not yet been confirmed. Caution may be warranted for specific patients, such as the elderly, and those with compromised immunity, short bowel syndrome, and cardiac valve disease [17]. To eliminate these risks, many studies on the functionality of substitutes, including heat-killed cells, cell components, and probiotic metabolites, have been actively conducted. Although probiotic intake is limited in cancer patients with significant immune deficiencies, few studies exist on its anti-cancer effects. 

*L. brevis* KU15176 was isolated from a traditional Korean food (cabbage kimchi) and has been reported to have probiotic activity [18]. This study investigated the effects of heat-killed *L. brevis* KU15176 and compared them with those of heat-killed *Lactobacillus rhamnosus* GG (LGG) on cell proliferation and apoptosis in gastric and colorectal cancer cell lines. 

## 2. Results

### 2.1. Nitric Oxide (NO) Production in RAW 264.7 Cells

The inflammatory reduction effect of heat-killed *Lactobacillus* strains was assessed using the NO assay because of the association between inflammation and cancer [19]. Figure 1 shows that lipopolysaccharides (LPS) treatment increased NO production in murine macrophage (RAW 264.7) cells through inflammatory responses. Similar to LGG, which was used as the comparative strain, NO production was significantly reduced by treatment with *L. plantarum* KU15149, *L. brevis* KU15159, and *L. brevis* KU15176 at 9 log CFU/mL.

### 2.2. Cell Proliferation Assay

The cell viability and cytotoxicity of heat-killed *Lactobacillus* strains were measured against normal and cancer cells utilizing 3-(4,5-dimethylthiazol-2-yl)-2,5-diphenyltetrazolium bromide (MTT) assays (Table 1 and Table 2). The viability of human lung fibroblasts (MRC-5) cells treated with these samples was greater than 100%. Therefore, it can be concluded that these strains did not reduce the cell viability of MRC-5 cells and can be considered as nontoxic to normal cells. Contrastingly, heat-killed KU15149, KU15159, and KU15176 at 9 log CFU/mL induced an anti-proliferative effect in human stomach adenocarcinoma (AGS), DLD-1, and LoVo cells (˃5% cytotoxicity).

### 2.3. Microscopic Analysis

Figure 2 shows the cell morphologies (AGS, DLD-1, and LoVo) treated with Roswell Park Memorial Institute (RPMI) medium and 9 log CFU/mL of heat-killed *Lactobacillus* strains. Compared to the control group, cell proliferation was inhibited after treatment with the sample. Most cells treated with the sample showed morphology of necrosis, that is, swollen cytoplasm, breakage of cell membrane, and release of cell contents. In contrast, AGS cells treated with KU15176 showed blebbing, a feature of apoptosis.

### 2.4. RNA Expression Using Semi-Quantitative Real-Time PCR

Real-time reverse transcription-polymerase chain reaction (RT-PCR) data showed that heat-killed *Lactobacillus* strains could regulate the expression amounts of some apoptotic genes. In comparison with the control, treatment of AGS cells with KU15176 caused a significant increase in *caspase-3* and *caspase-9* amounts (Figure 3). In addition, increased expression of Bcl-2 associated X-protein (*Bax*) was observed with 9 log CFU/mL treatment. However, there was minimal difference in the expression of B-cell lymphoma-2 (*Bcl*-2), an anti-apoptotic gene. Regarding the *Bax/Bcl-2* ratio, a measure of apoptosis sensitivity, KU15176 (106.74) also showed higher values than KU15149 (20.29) and KU15159 (70.79). However, in DLD-1 and LoVo cells treated with the sample, the expression of apoptotic genes was not affected (data not shown). KU15176 showed increases in *caspase-3* and *caspase-9* (2.59 and 1.98 times) expression. From these results, it is presumed that the cytotoxic effect of the sample, without the case of AGS cells treated with KU15176, was necrosis rather than apoptosis.

### 2.5. DAPI Staining

Microscopic analysis of cells stained through DAPI was used to examine the induction of apoptosis in AGS cells. Condensation or damage of the nucleus of cells, which is a representative result of apoptosis, was observed through DAPI staining to determine the presence of apoptosis [20]. Figure 4 shows that the morphology of the control group was normal, whereas AGS cells treated with KU15176 exhibited changes in morphological features, including condensed chromatin and nuclear fragmentation.

### 2.6. Flow Cytometry Analyses

Flow cytometry analysis showed the properties of the apoptotic cells. As shown in Figure 5, more than 94% of the cells of the control group were living (Figure 5A, LL). The groups of cells in treatment of 9 log CFU/mL of LGG and KU15149 showed more than 94% of living cells (Figure 5B, LL and Figure 5C, LL). Compared with these groups, the distribution of cells treated with KU15176 shifted to R4. The proportion of apoptotic cells dramatically increased to 19.24% (Figure 5E, UR + LR). Interestingly, cells treated with KU15159 showed relatively high apoptosis rates of 11.46% (Figure 5D, UR + LR), but higher necrosis rates (Figure 5C, UL) compared to KU15176.

### 2.7. Caspase Colorimetric Assay

Based on the central dogma, a colorimetric assay was conducted to detect changes in caspase-3 and caspase-9 activities during translation. The caspase-3 and capase-9 activity in the AGS cell line treated with KU15176 (9 log CFU/mL) were increased by approximately 3.1 and 2.1 times, respectively (Figure 6). Additionally, significant increase in caspase activity was observed in AGS cell line treated with KU15149 and KU15159, although smaller than the effect of KU15176.

## 3. Discussion

Currently, probiotics are being studied in many ways, and their various functions have been reported. Considering the increasing incidence of cancer in modern society, their anti-cancer effect is even more attractive. Probiotics can affect the expression of genes related to the cell cycle, which is essential for cancer treatment [21]. To avoid health problems that may occur due to the lowered immunity of cancer patients, it is hypothesized that the heat-killed method is more applicable.

Apoptosis is known to be caused by both intrinsic and extrinsic pathways. Signaling in both types of pathways activates caspases and cysteine proteases to eliminate dead cells. The difference between them is whether the stress that causes apoptosis originates from the outside or inside [22]. Therefore, intrinsic apoptosis is caused by intrinsic stresses such as oncogenes or DNA damage. The Bcl-2 family regulates the intrinsic apoptosis pathway [23]. Some Bcl-2 family proteins can function as pro-apoptotic proteins, such as *Bax* and *Bak*, whereas *Bcl-2* can function as an anti-apoptotic protein [24]. When an internal signal is detected, such as DNA damage, *Bax* and *Bak* are activated to form pores in the outer mitochondrial membrane, which interrupts the mitochondrial membrane potential and induces the release of cytochrome C from the inner mitochondrial membrane [25,26]. The released cytochrome C then binds to apoptotic protease activating factor 1 (APAF1), and this complex induces the activation of caspase-9. Activated caspase-9 activates other executable caspases, such as caspase-3, resulting in apoptosis.

KU15149 and KU15176 were found to be tolerant toward gastric acid and bile salts, intestinal adherence, and antibiotic sensitivity in previous experiments [18], and KU15159 also showed more than 98% acid and bile resistance (data not shown). Thus, these strains have been investigated for anti-cancer therapy. A series of experiments revealed that the anti-proliferative effect of KU15176 on AGS cells was due to apoptosis rather than necrosis. Figure 3 shows the upregulation of pro-apoptotic genes (*Bax*, *caspase-3*, and *caspase-9*) after treatment with heat-killed KU15176 (9 log CFU/mL). This can explain the process by which intrinsic apoptosis occurs owing to the increased expression of *Bax*, which belongs to the Bcl-2 gene family, leading to the activation of caspase-9 and caspase-3. Additionally, DNA damage and an apoptosis rate of 19.24% were observed at the same concentration. Caspase activity also increased 3.12 and 2.12 times for caspase-3 and caspase-9 compared to the control, respectively. Compared with LGG, which was used as a comparative strain, KU15176 exhibited promising effects.

Interestingly, the apoptosis rate was low (2.35%, data not shown), despite the increased expression of *caspase-3* and *caspase-9* at 8 log CFU/mL. This was presumed to be related to the change in the expression of *Bax*, which was different from the 9 log CFU/mL. The results of this experiment are presumed to differ greatly depending on the dose and treatment time of the sample, considering the lower caspase activity in the caspase colorimetric assay, which was executed after 24 h of sample treatment. Likewise, 9 log CFU/mL of the KU15159 treatment group was noteworthy. Effective caspase activity was observed in AGS cells treated with 9 log CFU/mL KU15159. Considering the high apoptosis rate (11.46%) despite the low expression amounts of caspase in RT-PCR, it seems that apoptosis induced by KU15159 can be terminated between 24 and 44 h. Likewise, the expression of *Bax*, *caspase-3*, and *caspase-9* in AGS cell line treated with KU15149 for 44 h was low, but caspase activity increasing. It is expected that the pro-apoptotic-genes-inducing effect of KU15149 was not observed due to the problem of sample treatment time. However, taking the low apoptosis rate into account, it is difficult to expect effective apoptosis occurrence. Considering the low selectivity of modern chemotherapy for cancer [27], the effect of KU15176, which induces apoptosis only in cancer cells, is impressive. Furthermore, KU15176 must be tested in vivo to verify its effectiveness.

## 4. Materials and Methods

### 4.1. Culture Media and Reagents

Lactobacilli MRS broth (BD Biosciences, Franklin Lakes, NJ, USA) was used for bacteria culture. RPMI 1640 medium, DMEM, penicillin/streptomycin (P/S), fetal bovine serum (FBS), and phosphate-buffered saline (PBS) were acquired from HyClone (Logan, UT, USA). All other reagents were obtained from Sigma-Aldrich (St. Louis, MO, USA).

### 4.2. Bacteria, Incubation Conditions, and Sample Preparation

*L. plantarum* KU15149, *L. brevis* KU15159, and *L. brevis* KU15176 were obtained from homemade diced radish and cabbage kimchi [18]. LGG was obtained from the Korean Collection for Type Cultures (KCTC, Seoul, Korea) and used as a reference strain. All bacteria were cultured in MRS broth at 37 °C for 18 h. The cultures were centrifuged at 12,000× *g* for 5 min at 4 °C, washed twice, and resuspended in PBS. The bacterial cells were heat-treated at 80 °C for 30 min in a water bath, centrifuged at 12,000× *g* for 5 min at 4 °C, and resuspended in cell growth media. Cell growth media without bacteria was used as a negative control.

### 4.3. Normal and Cancer Cell Lines and Culture Condition

MRC-5 (KCLB 10171), AGS (KCLB 21739), Caco-2 (KCLB 30037.1), DLD-1 (KCLB 10221), HT-29 (KCLB 30038), LoVo (KCLB 10229), and RAW 264.7 (KCLB 40071) cells were obtained from the Korean Cell Line Bank (KCLB, Seoul, Korea) and cultured in RPMI-1640 medium or DMEM, respectively. Each medium was supplemented with 1% P/S (*v*/*v*) and 10% FBS (*v*/*v*) and the cultures were incubated at 37 °C with 5% CO_2_.

### 4.4. Nitric Oxide Production in RAW 264.7 Cells

NO assays were accomplished to assess immunomodulatory potential [28]. RAW 264.7 cells were inoculated in a 96-well plate at a density of 2 × 10^5^ cells/well (100 μL) and incubated for 2 h. A total of 50 μL of diluted samples with the equal volume of 4 μg/mL LPS were added and LPS was used as a control for NO production. After 24 h incubation, 100 μL of the cell supernatant was blended with 100 μL of Griess reagent. The absorbance was assessed at 540 nm and determined applying the standard curve of sodium nitrite.

### 4.5. Cell Proliferation Assay

Cell proliferation was analyzed using the MTT assay [29,30]. Cells were seeded in 96-well plates and incubated overnight. The cells were treated with the samples (8 and 9 log CFU/mL) and incubated for 44 h. Next, cells were washed twice with PBS, treated with 100 μL of MTT reagent (0.5 mg/mL), and incubated for 4 h. Then, the MTT reagent was removed, and 200 μL of dimethyl sulfoxide (DMSO) was added. The absorbance was assessed at 570 nm, and cell viability was determined as follows:$$\mathrm{Viability}\;(\%)=\;\frac{A_{sample}}{A_{control}}\times 100$$
$$\mathrm{Cytotoxicity}\;(\%)=\;\left(1-\frac{A_{sample}}{A_{control}}\right)\times 100$$
where *A_sample_* and *A_control_* are the absorbance of the treated sample and control (cell growth media), respectively.

### 4.6. Microscopic Analysis

Cells were seeded in 96-well plates, incubated overnight, and treated with the sample (9 log CFU/mL). After 44 h of incubation, the cells were washed with PBS to remove probiotics attached to the cells. The cells were then treated with 200 μL RPMI and observed using an optical microscope (magnification: ×200) [30].

### 4.7. RNA Extraction and Semi-Quantitative Real-Time PCR

The *β-actin* housekeeping gene was used as a control. Total RNA was isolated from cells using the RNeasy^®^ Mini Kit (QIAGEN, Hilden, Limburg, Germany), and cDNA was synthesized using the Revert Aid First Strand cDNA Synthesis Kit (Thermo Fisher Scientific, Waltham, MA, USA) according to the manufacturer’s instructions. The expression amounts of apoptosis-related genes (*Bax*, *Bcl-2*, *caspase-3,* and *caspase-9*) in AGS, DLD-1, and LoVo cells were determined using SYBR Green PCR Master mix with semi-quantitative real-time RT-PCR (PikoReal 96, Scientific Pierce, Waltham, MA, USA). The primers used are listed in Table 3 [31,32,33].

The PCR conditions were 94 °C for 2 min, followed by 35 cycles at 94 °C for 15 s, 55 °C for 30 s, annealing at 68 °C for 60 s, and a final extension at 72 °C for 5 min. The results were analyzed using the delta–delta *Cq* method. A melting curve was used to analyze reaction specificity.

### 4.8. DAPI Staining

AGS cells were cultured at a seeding density of 2 × 10^4^ cells in confocal dishes and incubated overnight. After 9 log CFU/mL of sample was added to the cells, cells were incubated for 44 h. Then, the cells were washed with DAPI working solution (1 μg/mL), covered with DAPI working solution, and incubated for 15 min. Finally, microscopic observations were performed using a super-resolution confocal laser scanning microscope (Carl Zeiss LSM 800, Oberkochen, Germany) [34].

### 4.9. Apoptosis Assay

Flow cytometric analysis was performed to monitor the occurrence of apoptosis or necrosis in cells treated with KU15176. For this purpose, the Dead Cell Apoptosis Kit (Thermo Fisher) was used, as described in the commercial manual. The cells treated with the sample were harvested, washed with PBS, resuspended in 1 × annexin binding buffer, and stained with annexin V-FITC and PI solution for 15 min. Subsequently, the stained cells were resuspended in 1 × annexin binding buffer and prepared for monitoring apoptosis [35].

FITC and PI were detected in the FL-1 and FL-2 channels using CytoFLEX (Beckman Coulter, Brea, CA, USA), respectively. Cells were divided into four categories: necrotic cells (Annexin V−/PI+, UL), late apoptotic cells (Annexin V+/PI+, LR), early apoptotic cells (Annexin V+/PI−, UR), and living cells (Annexin V−/PI−, LL) [36].

### 4.10. Caspase-3 and Caspase-9 Colorimetric Assay

A colorimetric assay was performed to measure the effect of KU15176 on caspase activity in AGS cells. The AGS cell line was treated with the sample for 24 h. The caspase-3/CPP32 colorimetric assay kit (BioVision, Milpitas, CA, USA) and caspase-9/CPP32 colorimetric assay kit (BioVision) were used, as described in the commercial manual. Caspase activity was expressed as the fold change in the control group [37].

### 4.11. Statistical Analysis

All tested data are represented as the mean and standard deviation of three replicates. One-way analysis of variance (ANOVA) was used to verify significant differences. The mean values were used on behalf of Duncan’s multiple range test for post-hoc verification (*p* < 0.05). For statistical analysis, software SPSS (Version 25, IBM Corp., Armonk, NY, USA) was used. Significant differences for each characteristic were indicated in different letters.

## 5. Conclusions

There is an increasing need for experiments using heat-killed probiotics to avoid health problems such as sepsis, which can occur when patients with reduced immunity consume living organisms. Because of the anti-cancer effects of probiotics, heat-killed *L. brevis* KU15176 was investigated to detect the induction of apoptosis in AGS cell lines. The results showed that heat-killed KU15176 exhibited a selective anti-proliferative effect on AGS cell lines by regulating apoptosis-related genes such as *Bax*, *caspase-3*, and *caspase-9*. KU15176 induced DNA breakage or condensation in AGS cells and showed an apoptosis rate of 19.24 %. Collectively, it is concluded that KU15176 has anti-cancer potential in vitro, but it is necessary to check whether it is actually effective in in vitro tests and clinical trials. In this study, the anti-cancer effect according to the capacity and treatment time of KU15176 was different, so it is thought that the in vivo testing should be designed under various conditions.

## Figures and Tables

**Figure 1 ijms-23-04073-f001:**
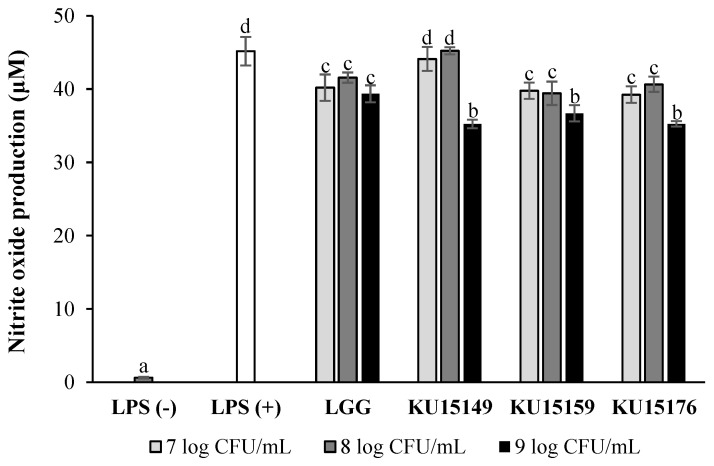
Anti-inflammatory effect of heat-killed *Lactobacillus* strains by nitrite oxide (NO) assay. Lipopolysaccharides (LPS [−]), negative control (DMEM); LPS (+), positive control (LPS, 1 μg/mL); LGG, *L. ramnosus* GG; KU15149, *L. plantarum* KU15149; KU15159, *L. brevis* KU15159; KU15176, *L. brevis* KU15176. LGG, KU15149, KU15159, and KU15176 were added to LPS (1 μg/mL). The data of NO production are the mean ± SE from three independent experiments. Different letters above value mean significant differences for each characteristic (*p* < 0.05).

**Figure 2 ijms-23-04073-f002:**
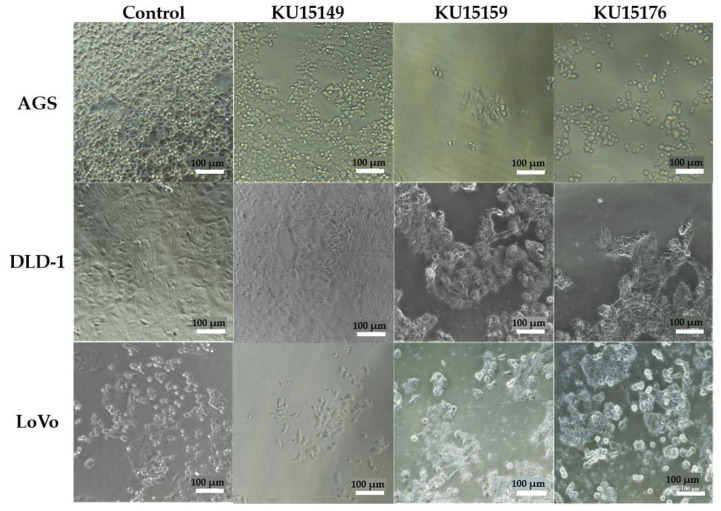
The morphology of cells after treated with Roswell Park Memorial Institute (RPMI) medium and 9 log CFU/mL of heat-killed *Lactobacillus* strains viewed using an optical microscope (magnification: ×200, white scale bar means 100 μm).

**Figure 3 ijms-23-04073-f003:**
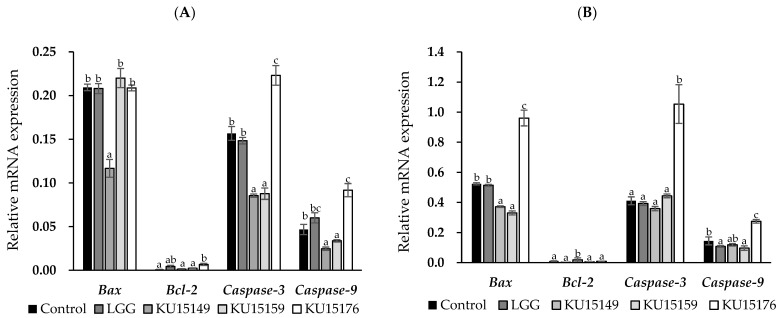
The expression of apoptosis-associated genes (*Bax*, *Bcl-2*, *caspase-3*, and *caspase-9*) of AGS cell line by heat-killed *Lactobacillus* strains using RT-PCR. (**A**) Treatment with 8 log CFU/mL, (**B**) Treatment with 9 log CFU/mL. The data of relative mRNA expression are represented as the mean ± SD of three experiments. The data of relative mRNA expression are the mean ± SE from three independent experiments. Different letters above value mean significant differences for each characteristic (*p* < 0.05).

**Figure 4 ijms-23-04073-f004:**
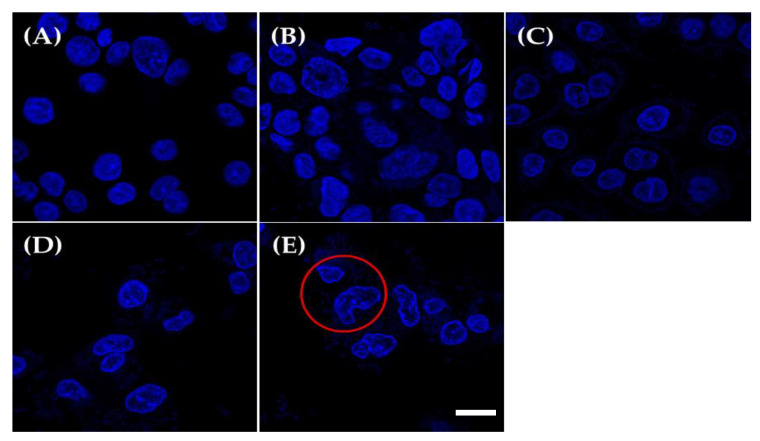
The morphological changes of heat-killed *Lactobacillus* strain-treated (9 log CFU/mL) AGS cells were determined using confocal imaging. (**A**) Roswell Park Memorial Institute (RPMI) medium (control), (**B**) LGG, (**C**) KU15149, (**D**) KU15159, and (**E**) KU15176. The red circle shows the condensation or breakage of the nucleus of treated cells; white scale bar means 10 μm.

**Figure 5 ijms-23-04073-f005:**
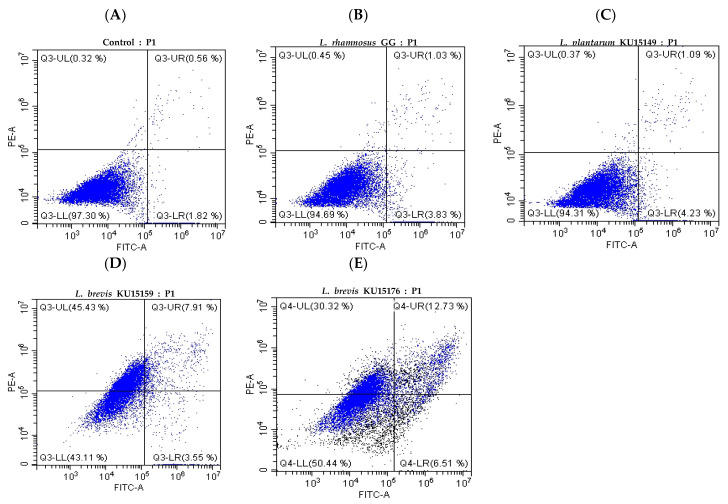
Apoptosis of AGS cells determined using flow cytometry after treatment with Roswell Park Memorial Institute (RPMI) (control) and 9 log CFU/mL of *Lactobacillus* strains. (**A**) RPMI (control), (**B**) LGG, (**C**) KU15149, (**D**) KU15159, and (**E**) KU15176. The cell scattering was analyzed using Annexin V-FITC and PI uptake.

**Figure 6 ijms-23-04073-f006:**
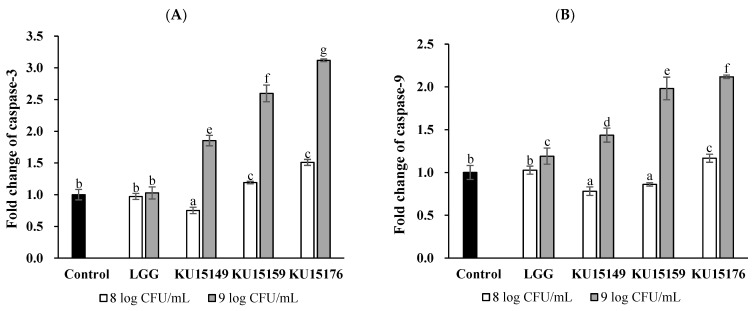
Activity of caspase-3 and caspase-9 was determined using fold change and the treated cells were compared with control. (**A**) caspase-3 activity, (**B**) caspase-9 activity The data of fold change of caspase-3 and caspase-9 are the mean ± SE from three independent experiments. Different letters above value mean significant differences for each characteristic (*p* < 0.05).

**Table 1 ijms-23-04073-t001:** Cell viability of MRC-5 cells by heat-killed cells of *Lactobacillus* strains using MTT assay.

LAB	Viability (%) of MRC-5 ^a^			
	LGG	*L. plantarum* KU15149	*L. brevis* KU15159	*L. brevis* KU15176
8 log CFU/mL	102.60 ± 2.18	113.04 ± 6.29	110.74 ± 2.06	124.54 ± 4.88
9 log CFU/mL	113.19 ± 2.24	112.99 ± 5.76	102.85 ± 3.04	127.37 ± 1.59

^a^ The data of cell viability are the mean ± SE from three independent experiments. CFU—colony-forming unit.

**Table 2 ijms-23-04073-t002:** Cytotoxicity against cancer cells of heat-killed cells of *Lactobacillus* strains using MTT assay.

LAB	Cytotoxicity (%) ^a^							
	LGG		*L. plantarum* KU15149		*L. brevis* KU15159		*L. brevis* KU15176	
	8 log CFU/mL	9 log CFU/mL	8 log CFU/mL	9 log CFU/mL	8 log CFU/mL	9 log CFU/mL	8 log CFU/mL	9 log CFU/mL
AGS	9.21 ± 1.47	31.94 ± 1.22	8.54 ± 0.61	35.52 ± 0.66	21.21 ± 0.62	37.60 ± 0.83	15.39 ± 0.80	41.34 ± 1.36
HT-29	1.12 ± 0.60	1.34 ± 1.23	0.90 ± 0.18	7.36 ± 1.20	1.31 ± 0.19	3.19 ± 0.45	1.67 ± 1.27	3.32 ± 0.95
DLD-1	49.37 ± 1.12	51.56 ± 3.02	37.28 ± 1.86	57.61 ± 3.61	45.85 ± 0.81	54.43 ± 0.93	42.22 ± 2.89	51.79 ± 3.10
LoVo	43.06 ± 1.94	48.14 ± 3.81	41.49 ± 4.84	61.34 ± 2.58	75.00 ± 2.22	76.00 ± 1.94	73.08 ± 3.35	69.56 ± 1.78
Caco-2	9.41 ± 0.16	17.49 ± 0.80	17.06 ± 3.45	25.30 ± 1.18	13.25 ± 0.46	26.22 ± 3.12	11.47 ± 2.22	27.33 ± 1.35

^a^ The data of cytotoxicity are the mean ± SE from three independent experiments. CFU—colony-forming unit.

**Table 3 ijms-23-04073-t003:** List of primer sequences used for semi-quantitative RT-PCR.

Primer		Sequence (5′ to 3′)
*β-Actin*	(Forward)	TTCTGACGGCAACTTCAACT
	(Reverse)	GTCCAGCCCATGATGGTTCT
*Bax*	(Forward)	TCACCCTGAAGTACCCCATC
	(Reverse)	GTCCAGCCCATGATGGTTCT
*Bcl-2*	(Forward)	CAGCTGCACCTGACGCCCTT
	(Reverse)	GCCTCCGTTATCCTGGATCC
*Caspase-3*	(Forward)	TTTGTTTGTGTGCTTCTGAGCC
	(Reverse)	ATTCTGTTGCCACCTTTCGG
*Caspase-9*	(Forward)	TGCTGCGTGGTGGTCATTCTC
	(Reverse)	CCGACACAGGGCATCCATCTG

RT-PCR—real-time PCR; *Bax*—Bcl-2-associated X protein; *Bcl-2*—B-cell lymphoma.

## Data Availability

The data of this study are availability from the corresponding author.

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
