# Peer review of "The Anti-Cancer Potential of Heat-Killed Lactobacillus brevis KU15176 upon AGS Cell Lines through Intrinsic Apoptosis Pathway"

_ijms, 2022, doi:10.3390/ijms23084073_

Round 1

Reviewer 1 Report

The present manuscript shows the effects of different heat-inactivated bacteria on different cell lines in order to demonstrate their anti-cancer effects. The results might be of interest for the audience to read and to take into consideration for future studies. I think that the manuscript could be considered for publication after minor revision:

  • The meaning of the letters a-d which appear in the graphs should be included in the figure legends.
  • The number of replicates for the experiment is missing in some figure legends such as Figure 1, Figure 5 and Figure 6. That should be included.
  • In figure legend 1 KU15159 and KU15176 are described as L. plantarum while they are L brevis in the text so this should be amended.
  • Figure 2 would be easier to follow if the authors include the bacteria or control media added to each cell culture in the top of the images and the different cell lines used in the left part of the panel.
  • In graph A of Figure 3 some bacteria reduce some molecules such as Bax and Caspase 3. This should be stated in the text and an explanation should be provided.
  • Regarding to Figure 5, the text talks about R1, R2, R3 and R4 while the panels in the figure are named UL, UR, LL and LR. This should be consistent between the text and the figure to make it easier to follow.
  • In Figure 6 only the increase with KU15176 is mentioned in the results section while the increase with KU15159 or with KU15149 is not mentioned. The authors should mention that as a result and, in the discussion, although they mention this fact for KU15159, they should explain why this happens with KU15149 too.

Author Response

Dear reviewer

We are appreciated for your comments improving quality of our manuscript. We followed the revision checklist for authors. Please confirm the response and revised manuscript: ijms-1645112_R1.

We hope the manuscript is now ready for the publication.

Thank you.

Hyun-Dong Paik

Department of Food Science and Biotechnology of Animal Resources

Konkuk University, Seoul 05029, Republic of Korea

Email address: [email protected]

Reviewers' Comments:

The present manuscript shows the effects of different heat-inactivated bacteria on different cell lines in order to demonstrate their anti-cancer effects. The results might be of interest for the audience to read and to take into consideration for future studies. I think that the manuscript could be considered for publication after minor revision:

  • The meaning of the letters a-d which appear in the graphs should be included in the figure legends.

- We thank you for pointing this out.

We have made some modifications to add the meaning of the letters (a-d) to the legend of figures which are analyzed by Duncan’s.

  • The number of replicates for the experiment is missing in some figure legends such as Figure 1, Figure 5 and Figure 6. That should be included.

- As your comment, we added the number of replicates for experiment in Figure 1 and Figure 6’s legends. However, in Figure 5, due to the characteristics of the flow cell analysis results classifying the distribution of cells on one screen, the only one time of experiment was conducted. We confirmed similar trends in repetitive experiments and selected the most distinct pictures among them and included them.

  • In figure legend 1 KU15159 and KU15176 are described as L. plantarum while they are L brevis in the text so this should be amended.

- Thanks for your comments for the mistake regarding the description of the lactic acid strain. We revised Figure 1 legend according to your comment.

  • Figure 2 would be easier to follow if the authors include the bacteria or control media added to each cell culture in the top of the images and the different cell lines used in the left part of the panel.

- Thank you for your positive feedback on the revision. Considering the length of the figure, we modified Figure 2 by adding the bacteria strain and control in the top of the image and the different cell lines in the left part of the image.

  • In graph A of Figure 3 some bacteria reduce some molecules such as Bax and Caspase 3. This should be stated in the text and an explanation should be provided.

- We apologize for the insufficient description. In the Result section, the decrease of Bax in AGS cell lines treated with KU15149 and KU15159 was introduced in result 4 through the concept of "Bax/Bcl-2". In the Discussion section, there was the explanation for the decrease in the expression of Bax and caspase in AGS treated with KU15159, but the explanation for KU15149 was not. So, an additional portion of the decrease in expression of Bax and caspase in AGS treated with KU15149 was added below the description (Line 150).

  • Regarding to Figure 5, the text talks about R1, R2, R3 and R4 while the panels in the figure are named UL, UR, LL and LR. This should be consistent between the text and the figure to make it easier to follow.

- Thank you for the comment. We made corrections on mis-region notation in Material, method, and Results (Line 136, 138, 295).

  • In Figure 6 only the increase with KU15176 is mentioned in the results section while the increase with KU15159 or with KU15149 is not mentioned. The authors should mention that as a result and, in the discussion, although they mention this fact for KU15159, they should explain why this happens with KU15149 too.

- Thank you for pointing this out. In addition to what you pointed out above, I added the explanation of effects of KU15149 on the discussion (Line 201).

Reviewer 2 Report

Review report on the manuscript titled The Anti-Cancer Effect of Heat-Killed Lactobacillus brevis KU15176 upon AGS Cell Lines through Intrinsic Apoptosis Pathway

Manuscript ID ijms-1645112

The manuscript has focused on the anti-cancer properties of Lactobacillus strains isolated from fermented foods. The experimental protocol was aimed to determine the relationship between inflammation and cancer, by assessing the anti-inflammatory effect of Lactobacillus brevis KU15176 using nitric oxide assay, while MTT assay was conducted to select cancer cells that showed the anti-proliferative effect of KU15176.  RT-PCR, DAPI staining, flow cytometry, and caspase colorimetric assay were performed, suggesting a potential prophylactic effect of Lactobacillus brevis KU15176 against cancer.

The manuscript is well organized and well written, suitable for publication in IJMS.

However, I still have some comments that are necessary to in order to highlight the results and the importance of the study.

  1. In the introduction section the authors mentioned that probiotics exhibit functional effects, such as anti-inflammatory, antioxidant, immune-enhancing, antimicrobial and

antidiabetic activities. I would like to suggest another effect of Lactic acid bacteria (LAB) recently demonstrated, being successfully used in the removal of heavy metal (lead, cadmium) from different environments and protective effect against cadmium toxicity (https://doi.org/10.1016/j.molstruc.2021.131325). These aspects are useful to be mentioned in the introduction section of the manuscript.

  1. In figure 2 please include additional explanation about the notation A1, A2,…B1,B2, C1,C2…C4. Otherwise it is difficult to interpret the significance of the images, being confusing. In the same time, please provide a more clearly scale bar for each image.
  2. Please provide scale bar in the images presented in Figure 4.
  3. As the discussion related to Figure 5 refers to R1, R2, R3 and R4, please provide labels for each one, otherwise is confusing.
  4. The conclusion section should provide more details related to further approaches.

In conclusion, I recommend the publication, after revision taking into account the above comments.

Author Response

Dear reviewer

We are appreciated for your comments improving quality of our manuscript. We followed the revision checklist for authors. Please confirm the response and revised manuscript: ijms-1645112_R1.

We hope the manuscript is now ready for the publication.

Thank you.

Hyun-Dong Paik

Department of Food Science and Biotechnology of Animal Resources

Konkuk University, Seoul 05029, Republic of Korea

Email address: [email protected]

Reviewers' Comments:

Review report on the manuscript titled The Anti-Cancer Effect of Heat-Killed Lactobacillus brevis KU15176 upon AGS Cell Lines through Intrinsic Apoptosis Pathway

Manuscript ID ijms-1645112

The manuscript has focused on the anti-cancer properties of Lactobacillus strains isolated from fermented foods. The experimental protocol was aimed to determine the relationship between inflammation and cancer, by assessing the anti-inflammatory effect of Lactobacillus brevis KU15176 using nitric oxide assay, while MTT assay was conducted to select cancer cells that showed the anti-proliferative effect of KU15176.  RT-PCR, DAPI staining, flow cytometry, and caspase colorimetric assay were performed, suggesting a potential prophylactic effect of Lactobacillus brevis KU15176 against cancer.

The manuscript is well organized and well written, suitable for publication in IJMS.

However, I still have some comments that are necessary to in order to highlight the results and the importance of the study.

  1. In the introduction section the authors mentioned that probiotics exhibit functional effects, such as anti-inflammatory, antioxidant, immune-enhancing, antimicrobial and antidiabetic activities. I would like to suggest another effect of Lactic acid bacteria (LAB) recently demonstrated, being successfully used in the removal of heavy metal (lead, cadmium) from different environments and protective effect against cadmium toxicity (https://doi.org/10.1016/j.molstruc.2021.131325). These aspects are useful to be mentioned in the introduction section of the manuscript.

- At first, we appreciate your advice for further strengthening the explanation of the introduction. We added the effect you suggested to the effect of lactic acid bacteria and cited the paper (Line 29).

  1. In figure 2 please include additional explanation about the notation A1, A2,…B1,B2, C1,C2…C4. Otherwise it is difficult to interpret the significance of the images, being confusing. In the same time, please provide a more clearly scale bar for each image.

- Thank you for your positive feedback on the revision. Considering the length of the figure, we modified Figure 2 by adding the bacteria strain and control in the top of the image and the different cell lines in the left part of the image.

  1. Please provide scale bar in the images presented in Figure 4.

- We checked the part you mentioned, but Figure 4 already shows the scale bar. However, we judged that this was a problem caused by the small size of the picture, so we re-marked the larger scale bar and added an explanation to the legend.

  1. As the discussion related to Figure 5 refers to R1, R2, R3 and R4, please provide labels for each one, otherwise is confusing.

- Thank you for the comment. To prevent confusion, we modified the name of region to LL, UL, LU, and UU used in Figure 5.

  1. The conclusion section should provide more details related to further approaches.

- We agree with what you pointed out. So, we added more details about the approach of the in vivo experiment to the conclusion (Line 320).

In conclusion, I recommend the publication, after revision taking into account the above comments.

Reviewer 3 Report

The manuscript entitled: “The Anti-Cancer Effect of Heat-Killed Lactobacillus brevis KU15176 upon AGS Cell Lines through Intrinsic Apoptosis Pathway” should be better assessed regarding the limits and reached end points expexted by the Authors, exploiing better the Conclusion section. There are still a few remar sas dtailed in the following. The Authors state that: “L. plantarum KU15149, L. brevis KU15159, and L. brevis KU15176 were obtained from homemade diced radish and cabbage kimchi”: some more details should be given on this point: please add data and justify why these have been chosen for the evaluation. The cell lines used require an etical commitee authorization? Please comment on this point. At line 197 the Authors mention “selectivity”: please substantiate this point which is interesting and shold be better commented. Please exploit better the Conclusions ection with reference to end points and limits of the proposed manuscript.

Author Response

Dear reviewer

We are appreciated for your comments improving quality of our manuscript. We followed the revision checklist for authors. Please confirm the response and revised manuscript: ijms-1645112_R2.

We hope the manuscript is now ready for the publication.

Thank you.

Hyun-Dong Paik

Department of Food Science and Biotechnology of Animal Resources

Konkuk University, Seoul 05029, Republic of Korea

Email address: [email protected]

Reviewers' Comments:

The manuscript entitled: “The Anti-Cancer Effect of Heat-Killed Lactobacillus brevis KU15176 upon AGS Cell Lines through Intrinsic Apoptosis Pathway” should be better assessed regarding the limits and reached end points expected by the Authors, exploring better the Conclusion section.

--> Thank you for your comments on the correction. First of all, we revised the title as “Anti-Cancer Potential” according to your advice.

There are still a few remar sas detailed in the following. The Authors state that: “L. plantarum KU15149, L. brevis KU15159, and L. brevis KU15176 were obtained from homemade diced radish and cabbage kimchi”: some more details should be given on this point: please add data and justify why these have been chosen for the evaluation.

--> The reason for the choice of these three strains (KU15149, KU15159, and KU15176) is explained in a series of processes in the text. The results of NO assay and MTT assay confirmed that these three strains were more effective than other strains, so only the data of the strain was adopted in the paper. We suggested in Line 180-184.

The cell lines used require an etical commitee authorization? Please comment on this point. At line 197 the Authors mention “selectivity”: please substantiate this point which is interesting and should be better commented. Please exploit better the Conclusions section with reference to end points and limits of the proposed manuscript.

--> We would like to inform you that the "etical commitee authorization" you mentioned is not applicable because we purchased cell lines from Korean Cell Line Bank (KCLB, Seoul, Korea). In addition, as evidence of "selectivity" you pointed out, we added references to replace the explanation (Maleki, E.H.; Bahrami, A.R.; Sadeghian, H.; Matin, M.M. Discovering the structure–activity relationships of different O-prenylated coumarin derivatives as effective anticancer agents in human cervical cancer cells. Toxicol. In Vitro 202063, 104745). Finally, considering the limitations of this study, the necessity of clinical trials was added to the results (in Line 322-325).

Reviewer 4 Report

I carefully read the manuscript by Dr. Paik et al. that could be of potential interest for the readers of the journal.

My comments and suggestions for the authors are the following:

  • English language needs to be carefully revised and improved
  • Tables: All the abbreviations used should be defined at the bottom of the table, as per the Instructions for the Authors of the Journal
  • Statistical methods are roughly described.
  • The authors should deeply discuss the limitations of their study.
  • In the conclusion section (either in the manuscript and in the abstract), the authors should specify that their findings are preliminary and evidence in human are necessary to confirm their observations.

Author Response

Dear reviewer

We are appreciated for your comments improving quality of our manuscript. We followed the revision checklist for authors. Please confirm the response and revised manuscript: ijms-1645112_R2.

We hope the manuscript is now ready for the publication.

Thank you.

Hyun-Dong Paik

Department of Food Science and Biotechnology of Animal Resources

Konkuk University, Seoul 05029, Republic of Korea

Email address: [email protected]

Reviewers' Comments:

I carefully read the manuscript by Dr. Paik et al. that could be of potential interest for the readers of the journal.

à Thank you for your thoughtful comment. Following your advice, we added the definition of abbreviation below the table.

My comments and suggestions for the authors are the following:

English language needs to be carefully revised and improved.

--> We edited by editing system and adjusted as similarity by ithenticate.

Tables: All the abbreviations used should be defined at the bottom of the table, as per the Instructions for the Authors of the Journal.

--> Following your advice, we added the definition of abbreviation below the table.

Statistical methods are roughly described.

--> As your comments, the statistical method also added more explanation to make it more detailed.

The authors should deeply discuss the limitations of their study.

In the conclusion section (either in the manuscript and in the abstract), the authors should specify that their findings are preliminary and evidence in human are necessary to confirm their observations.

--> Finally, in consideration of the limitations of this study, a mention of the necessity of “clinical trials” was added to the result.

Reviewer 5 Report

This study assesses the anticancer effect of Lactobacillus brevis, which could exert an anti-inflammatory action on tumor cells and induces apoptosis. Several assays were performed to evaluate the antiproliferative and anti-inflammatory effects, and the expression of various apoptosis-associated genes (Bax, Bcl-2, caspase-3, caspase-9). The manuscript contains five keywords, six figures, three tables, and thirty-six references. Overall, it is a correct, complete, and well-conducted article.

Supplementary comments on different sections of the manuscript are also made.

Keywords
The manuscript shows five keywords. For keywords, where possible, please use Medical Subject Headings Terms (MeSH Terms). “Anti-cancer effect” is not a MeSH term. Alternative MeSH terms proposed could be: “antineoplastic agents” better than “anti-cancer effect", and “caspases”, or “caspase-3” and “caspase-9”, rather than “caspase”.
These suggestions about keywords are optional, not mandatory.

Introduction
Correct, no comments.

Results
Correct, no comments.

Discussion
Correct, no comments.

Materials and Methods
A complete and well-written section.

Conclusions
Correct, no comments.

Abbreviations
Fine, a very appropriate section.

References
Total number of manuscript references: 36.
A correct section. The reference format is according to the journal’s guidelines.

Figures
Total number of manuscript figures: 6

Tables
Total number of manuscript tables: 3
Consider including table footers explaining abbreviations, e.g. “CFU/mL”, “RT-PCR”.

Author Response

Dear reviewer

We are appreciated for your comments improving quality of our manuscript. We followed the revision checklist for authors. Please confirm the response and revised manuscript: ijms-1645112_R2.

We hope the manuscript is now ready for the publication.

Thank you.

Hyun-Dong Paik

Department of Food Science and Biotechnology of Animal Resources

Konkuk University, Seoul 05029, Republic of Korea

Email address: [email protected]

Reviewers' Comments:

This study assesses the anticancer effect of Lactobacillus brevis, which could exert an anti-inflammatory action on tumor cells and induces apoptosis. Several assays were performed to evaluate the antiproliferative and anti-inflammatory effects, and the expression of various apoptosis-associated genes (Bax, Bcl-2, caspase-3, caspase-9). The manuscript contains five keywords, six figures, three tables, and thirty-six references. Overall, it is a correct, complete, and well-conducted article.

Supplementary comments on different sections of the manuscript are also made.

--> Thank you for review and comments. We revised as your comments.

Keywords

The manuscript shows five keywords. For keywords, where possible, please use Medical Subject Headings Terms (MeSH Terms). “Anti-cancer effect” is not a MeSH term. Alternative MeSH terms proposed could be: “antineoplastic agents” better than “anti-cancer effect", and “caspases”, or “caspase-3” and “caspase-9”, rather than “caspase”.

--> Thank you for comments. In consideration of your advice, we have revised the keyword to Medical Subject Headings Terms. Then, we added explanations of the abbreviation you mentioned on the tables.

These suggestions about keywords are optional, not mandatory.

Introduction

Correct, no comments.

Results

Correct, no comments.

Discussion

Correct, no comments.

Materials and Methods

A complete and well-written section.

Conclusions

Correct, no comments.

Abbreviations

Fine, a very appropriate section.

References

Total number of manuscript references: 36.

A correct section. The reference format is according to the journal’s guidelines.

Figures

Total number of manuscript figures: 6

Tables

Total number of manuscript tables: 3

Consider including table footers explaining abbreviations, e.g. “CFU/mL”, “RT-PCR”.

--> We added explanations of the abbreviation you mentioned on the tables.

Round 2

Reviewer 4 Report

Dear Editor,

I carefully read the revised version of the manuscript that is improved in comparison with the previous version.